# Generalizing Unsupervised Anomaly Detection: Towards Unbiased Pathology Screening

**Cosmin I. Bercea**[1,2]                                    COSMIN.BERCEA@TUM.DE

**Benedikt Wiestler**[3]                                        B.WIESTLER@TUM.DE

**Daniel Rueckert**[1,3,4]                                  DANIEL.RUECKERT@TUM.DE

**Julia A. Schnabel**[1,2,5]                                JULIA.SCHNABEL@TUM.DE

[1] *Faculty of Informatics, Technical University of Munich (TUM), Munich, Germany*

[2] *Helmholtz AI and Helmholtz Center Munich, Germany*

[3] *Klinkum Rechts der Isar, Munich, Germany*

[4] *Department of Computing, Imperial College London, United Kingdom*

[5] *School of Biomedical Engineering and Imaging Sciences, King's College London, United Kingdom*

**Editors:** Accepted for publication at MIDL 2023

## Abstract

The main benefit of unsupervised anomaly detection is the ability to identify arbitrary instances of pathologies even in the absence of training labels or sufficient examples of the rare class(es). Even though much work has been done on using auto-encoders (AE) for anomaly detection, there are still two critical challenges to overcome: First, learning compact and detailed representations of the healthy distribution is cumbersome. Second, the majority of unsupervised algorithms are tailored to detect hyperintense lesions on FLAIR brain MR scans. We found that even state-of-the-art (SOTA) AEs fail to detect several classes of non-hyperintense anomalies on T1w brain MRIs, such as brain atrophy, edema, or resections. In this work, we propose *reversed AEs (RA)* to generate pseudo-healthy reconstructions and localize various brain pathologies. Our method outperformed SOTA methods on T1w brain MRIs, detecting more global anomalies (AUROC increased from 73.1 to 89.4) and local pathologies (detection rate increased from 52.6% to 86.0%).

**Keywords:** Unsupervised Anomaly Detection, Pathology Screening.

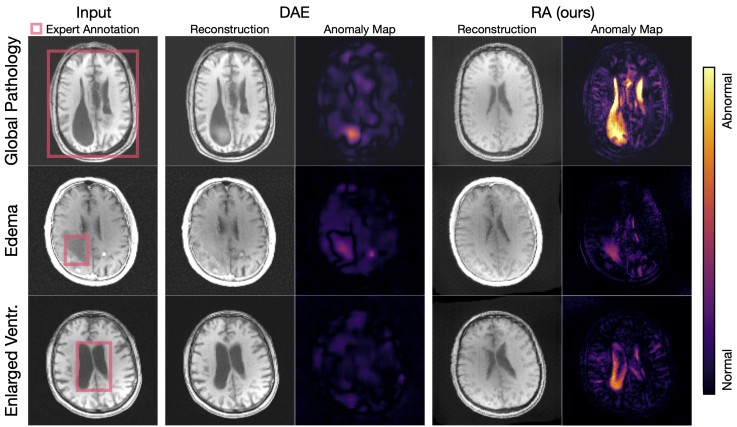

Figure 1: SOTA pathology detection methods, such as DAE (Kascenas et al., 2022) may miss structural anomalies like atrophy. In contrast, our proposed method *RA* generates pseudo-healthy reconstructions for unbiased detection of various diseases.

# 1. Introduction

The automated detection of abnormal findings is a critical component in the clinical workflow, enabling clinical screening, early disease detection and rapid triage of acute cases. Supervised machine learning methods (Kamnitsas et al., 2017; Vyas et al., 2018) use large amounts of expert-annotated datasets to target specific diseases, such as brain tumours or lesions. However, they generalize poorly beyond the trained distribution and learned diseases, due to the heterogeneity and large amount of existing pathologies (Ruff et al., 2021). In contrast, unsupervised anomaly detection aims to leverage the large amounts of available datasets containing normal samples to learn the distribution that best describes the healthy population. Thus, it is natural to detect any anomaly by identifying patterns that do not match the expected normative distribution. Reconstruction-based auto-encoders (AEs) are optimized to only reconstruct in-distribution (ID) samples well, and detect anomalies from imperfect reconstructions of abnormal data. Even though they have become a very popular framework for unsupervised anomaly detection and are widely used in medical imaging (Baur et al., 2021; Bercea et al., 2022; Gong et al., 2019; Pawlowski et al., 2018; Zimmerer et al., 2019a), there are still two important challenges to overcome.

First, it is difficult to learn concise and accurate representations of the healthy distribution. Recent work suggests that AEs might reconstruct out-of-distribution (OoD) samples even better than ID instances (Nalisnick et al., 2018; Ren et al., 2019; Schirrmeister et al., 2020), which renders them unsuitable for anomaly detection. Several techniques are used in the literature to regularize AEs, such as enforcing sparsity (Makhzani and Frey, 2013) or training proxy self-supervised tasks (Vincent et al., 2010; Zimmerer et al., 2019b). Recent de-noising auto-encoders (Kascenas et al., 2022) achieved SOTA brain tumour segmentation by learning to remove generated synthetic noise patterns. However, this is advantageous only when the anomaly distribution is known *a priori*. Variational AEs (VAEs) (Kingma and Welling, 2013; Zimmerer et al., 2019a) regularize the distribution over the latent space to be similar to a prior distribution, usually a standard isotropic Gaussian. While these methods capture the training distribution better, they are known to produce blurry reconstructions. In contrast, generative adversarial networks (GANs) (Goodfellow et al., 2014) can generate high-resolution images and have been applied to medical anomaly detection (Schlegl et al., 2019). More recently, Daniel and Tamar (2021) combined VAEs and GANs, by training the encoder and the decoder in an introspective way, achieving SOTA image generations and outlier detection performance. However, their reconstructions do not perfectly match the inputs, leading to high amount of false positive residual errors.

Second, the majority of the unsupervised algorithms are tailored to detect hyper-intense regions on FLAIR brain MRI. In Baur et al. (2021) most SOTA algorithms fail without hyper-intense specific post-processing, such as only keeping the positive residual errors. Furthermore, simple thresholding techniques have been recently shown to outperform unsupervised deep learning models for white lesion detection (Meissen et al., 2022a). To advance the field of unsupervised anomaly detection, it is therefore critical to develop methods that go beyond hyper-intensity thresholding to enable clinical screening and the detection and localization of multiple, different pathologies.

In this work, we investigate whether AEs learn meaningful representations to detect and localize multiple different pathologies and artefacts on T1w brain MRIs. First, we evaluate whether AEs can learn the healthy anatomy, i.e., absence of pathology, and produce pseudo-healthy reconstructions of normal brain scans that were augmented with synthetic anomalies. Second, we evaluate whether AEs can detect global and local anomalies on a challenging brain dataset, containing 16 different pathologies. We found that AEs fail at detecting multiple diseases by either reconstructing the anomalies, or producing blurry or inaccurate reconstructions. To address these issues, we propose *RA*, novel *reversed AEs* to

produce accurate pseudo-healthy reconstructions of abnormal samples and localize various pathologies. Figure 1 shows example results of $RA$ compared to SOTA DAE.

Our manuscript advances the field of unsupervised anomaly detection by paving the way towards generalized pathology localization. In summary, our contributions are:

- We quantify how well AEs can learn the nominal distribution and reconstruct pseudo-healthy images of abnormal brain scans using synthetic experiments.

- We evaluate SOTA AEs on a wide range of local non-hyperintense anomalies.

- We propose *reversed AEs*, *RA*, a novel method that considerably outperforms SOTA unsupervised methods on pathology detection and localization.

## 2. Background

**Unsupervised anomaly detection.** In the following we will refer to 'normal' as the absence of pathologies. Given a set of normal samples $x \in \mathcal{X} \subset \mathbb{R}^N$, the AE objective is to find a function $f : \mathbb{R}^N \rightarrow \mathbb{R}^D$ and its inverse $g : \mathbb{R}^D \rightarrow \mathbb{R}^N$, such that $x \approx g(f(x))$. Usually $f$ and $g$ are called encoder and decoder and the mapping $f$ projects the input to a much lower-dimensional representation. The core assumption of unsupervised anomaly detection is that the learned representations would contain features describing the nominal distribution even for outlier samples $\overline{x} \notin \mathcal{X}$. Thus, $x_{ph} = (g(f((\overline{x}))) \in \mathcal{X}$ would be the pseudo-healthy reconstruction of $\overline{x}$. An anomaly score is therefore usually derived directly from the pixel-wise difference between an input and its reconstruction: $s(x) = |x - g(f(x))|$.

**Evidence lower bound (ELBO).** From a variational inference setting (Kingma and Welling, 2013), our aim is to optimize the parameters $\theta$ of a latent variable model $p_\theta(\mathrm{x})$ by maximizing the log-likelihood, $\log p_\theta(x)$ of the observed samples $x$. The term is intractable, but we can approximate the true posterior $p_\theta(z|x)$ with a proposal distribution $q_\phi(z|x)$:

$$\log p_\theta(x) \geq \mathbb{E}_{q(z|x)}[\log p_\theta(x|z)] - KL[q_\phi(z|x)||p(z)] = ELBO(x) \tag{1}$$

Here KL denotes the Kullback-Leibler divergence; $q_\phi(z|x)$ and $p_\theta(x|z)$ are usually known as the encoder $E_\phi$ and decoder $D_\theta$, which are neural networks with parameters $\phi$ and $\theta$. For VAEs, the prior $p(z)$ is usually the normal distribution $\mathcal{N}(\mu_0, \sigma_0)$ and the ELBO can be maximized using the reparameterization trick.

**Soft-introspective VAEs (SI-VAE)** (Daniel and Tamar, 2021) added an adversarial loss to the VAE training. The intuition behind it is to combine the encoding capabilities of VAEs with the powerful image synthesis ability of GANs (Goodfellow et al., 2014). In contrast to Pidhorskyi et al. (2018), SI-VAEs do not add additional discriminator networks, but use the VAEs encoder and decoder in an adversarial way. The encoder is encouraged to differentiate between real and generated samples by minimizing the KL of the latent distribution of real samples and the prior and maximizing the KL of generated samples. In contrast, the decoder is trained to 'fool' the encoder by reconstructing real data samples using the standard ELBO and minimizing the KL of generated samples compressed by the encoder. The objectives of the encoder and decoder which are *to be maximized* are:

$$\mathcal{L}_{E_\phi}(x, z) = ELBO(x) - \frac{1}{\alpha}(exp(\alpha ELBO(D_\theta(z)), \tag{2}$$
$$\mathcal{L}_{D_\theta}(x, z) = ELBO(x) + \gamma ELBO(D_\theta(z)),$$

where $\alpha \geq 0$ and $\gamma \geq 0$ are hyper-parameters.

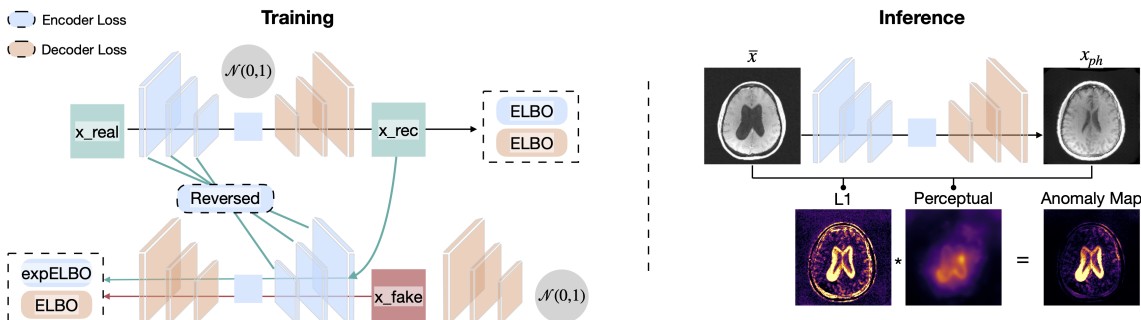

Figure 2: The *RA* architecture uses an encoder and decoder that optimize the ELBO for real samples. The reversed embedding loss improves the accuracy of feature representations, while expELBO and ELBO respectively encourage the encoder to push away generated samples from the latent space and the decoder to fool the encoder. At inference, *RA* generates pseudo-healthy reconstructions $(x_{ph})$ and produces anomaly maps by combining residual and perceptual differences.

## 3. RA: Reversed Autoencoders

SI-VAE reconstructions are imperfect, resulting in large false positive residuals that obscure errors of interest, i.e., pathologies. We propose new *reversed AEs* (see Figure 2) to mitigate these issues by incorporating a reversed multi-scale embedding loss into the encoder and computing the anomaly scores using residual and perceptual differences.

**Reversed embedding similarity.** We focus our proposed method on learning better latent representations by augmenting the encoder with a multi-scale embedding similarity loss. We therefore enforce that input representations match the embeddings of the generated reconstructions. This reverse comparison is performed at multiple levels, inspired by knowledge distillation methods (Salehi et al., 2021):

$$\mathcal{L}_{Reversed}(x) = \sum_{l=0}^{L}(1 - \mathcal{L}_{Sim}(E_\phi^l(x), E_\phi^l(x_{rec})) + \frac{1}{2}MSE(E_\phi^l(x), E_\phi^l(x_{rec})), \quad (3)$$

where $E_\phi^l$ is the $l$-th embedding of the $L$ encoder layers, $x_{rec} = D_\theta(E_\phi(x))$, $\mathcal{L}_{Sim}$ is the cosine similarity, and $MSE$ the mean squared error. The overall encoder objective is:

$$\mathcal{L}_{E_\phi}(x, z) = ELBO(x) - \frac{1}{\alpha}(exp(\alpha ELBO(D_\theta(z)) + \lambda\mathcal{L}_{Reversed}(x), \quad (4)$$

with $\lambda$ a weight that we empirically set to $5e-3$. The decoder loss remains as in Equation 2.

**Anomaly scores.** The most common approach for detecting anomalies is to calculate the residuals between anomalous inputs and their pseudo-healthy reconstructions. However, these errors have been shown to depend strongly on the underlying intensities (Meissen et al., 2022b). To address this issue we apply adaptive histogram equalization (eq) before computing the residual and augment the errors with perceptual differences for robustness:

$$s(x) = |(eq(x_{ph}) - eq(\overline{x})| * (\mathcal{S}_{lpips}(x_{ph}, \overline{x}) * \mathcal{S}_{lpips}(eq(x_{ph}), eq(\overline{x})), \quad (5)$$

with $\mathcal{S}_{lpips}$ being the learned perceptual image patch similarity (Zhang et al., 2018).

Table 1: Experiments with synthetic anomalies, i.e., sprites of different intensities and 'copy-paste' textural changes. See Figure 3 for visual results.

($a$) **Pseudo-healthy Reconstruction.** *RA* achieves the best mean squared error (MSE) and learned perceptual image patch similarity (LPIPS) (Zhang et al., 2018) on the synthetically augmented regions.

| Method | Intensity Anomaly MSE ↓ | Intensity Anomaly LPIPS ↓ | Texture Anomaly MSE ↓ | Texture Anomaly LPIPS ↓ |
|---|---|---|---|---|
| AE-S | $17.6 \pm 7.2$ | $29.3 \pm 10.0$ | $11.8 \pm 7.3$ | $28.8 \pm 11.7$ |
| VAE | $4.4 \pm 5.70$ | $44.3 \pm 6.6$ | $5.3 \pm 5.5$ | $41.9 \pm 8.7$ |
| DAE | $25.8 \pm 10.6$ | $32.1 \pm 9.4$ | $8.9 \pm 6.0$ | $25.1 \pm 12.1$ |
| SI-VAE | $3.3 \pm 3.2$ | $19.8 \pm 6.4$ | $5.3 \pm 4.8$ | $19.4 \pm 6.2$ |
| RA (ours) | $\mathbf{3.1 \pm 2.5}$ | $\mathbf{17.7 \pm 7.6}$ | $\mathbf{4.8 \pm 5.4}$ | $\mathbf{18.5 \pm 5.8}$ |

($b$) **Anomaly Segmentation.** *RA* achieves the best AUPRC for both synthetic anomaly types.

| Method | Int. Ano. AUPRC ↑ | Tex. Ano. AUPRC ↑ |
|---|---|---|
| AE-S | 5.8 | 7.9 |
| VAE | 24.2 | 20.0 |
| DAE | 57.9 | 30.1 |
| SI-VAE | 27.6 | 20.5 |
| RA (ours) | **73.7** | **57.4** |

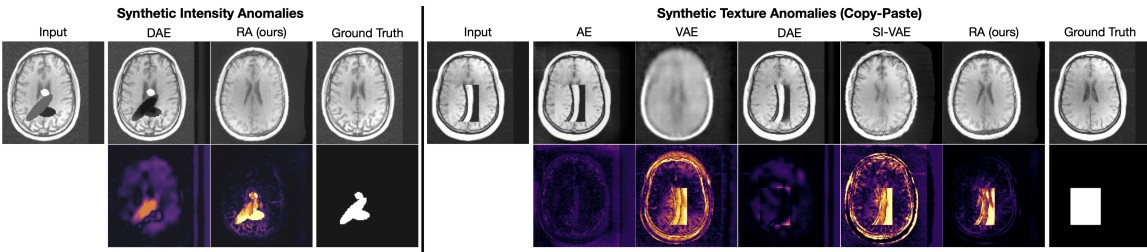

Figure 3: Synthetic experiments. *RA* removes intensity and textural anomalies and yields the most accurate pseudo-healthy reconstructions compared to the ground truth.

## 4. Experiments

**Datasets.** For an evaluation on clinical data, we use two different brain T1w MRI datasets: FastMRI+ (Zhao et al., 2021) contains healthy and pathological axial brain MRI scans with expert bounding box annotation for 30 different pathology categories. We selected the middle axial slice, which resulted in: 176 normal scans (131 train, 15 validation, and 30 test); 27 scans containing global pathologies; 23 global artefacts; and 171 local pathologies with expert bounding box annotations. We supplemented the training with 581 T1w healthy brain MRI middle axial slices from IXI (IXI, 2012) (see Appendix for details).

### 4.1. Representation learning for healthy distributions

In this section, we investigate whether AEs can learn compact and accurate representations of the healthy distribution. Therefore, we conduct two synthetic experiments with intensity and textural anomalies, as shown in Figure 3. We quantify the pseudo-healthy reconstruction performance of the methods in Table 1($a$) and the anomaly segmentation results in Table 1($b$) using the provided healthy images and synthetic masks as ground truth. AEs (Baur et al., 2021) copy anomalies, resulting in poor reconstruction and anomaly segmentation accuracy. VAEs (Zimmerer et al., 2019a) learn compact representations pre-

Table 2: Experiments on T1w brain MRIs with global diseases, artefacts, and 13 different local pathologies with bounding-box annotations. Figure 4(b) shows visual results.

(a) **Anomaly Detection.** $RA$ increases the area under the receiver-operator point (AUROC) by 32.4% for pathology detection and by 13.5% for detecting global artefacts.

| Method | Average | Global Pathology | Global Artefacts |
|---|---|---|---|
| | | AUROC ↑ | |
| AE | $47.4 \pm 7.6$ | $49.3 \pm 5.7$ | $45.6 \pm 9.6$ |
| VAE | $62.9 \pm 6.9$ | $62.0 \pm 7.0$ | $63.8 \pm 6.7$ |
| DAE | $57.9 \pm 10.0$ | $58.2 \pm 10.3$ | $57.6 \pm 9.6$ |
| SI-VAE | $73.1 \pm 5.6$ | $67.2 \pm 6.2$ | $79.1 \pm 5.0$ |
| MKD | $62.2 \pm 4.9$ | $61.3 \pm 6.5$ | $63.2 \pm 3.3$ |
| RA (ours) | $\mathbf{89.4 \pm 1.6}$ | $\mathbf{89.0 \pm 2.5}$ | $\mathbf{89.8 \pm 0.7}$ |

(b) **Pathology localization.** $RA$ is the most robust method ($\overline{F1}$), detecting 86% of the local pathologies.

| Method | Local Pathology | |
|---|---|---|
| | #Detected ↑ | $\overline{F1}$(%) ↑ |
| AE-S | 21/171 | 2.5 |
| VAE | 79/171 | 9.8 |
| DAE | 90/171 | 29.0 |
| SI-VAE | 82/171 | 9.9 |
| MKD | 83/171 | 26.5 |
| RA (ours) | **147/171** | **37.9** |

venting the reconstruction of anomalies (low MSE score) but produce blurry results (high LPIPS score). The resulting false positives yield poor segmentation performance. Interestingly, while DAE (Kascenas et al., 2022) achieves very poor reconstruction results, it outperforms the other baselines in segmenting the anomalies. It does so by in-painting structures that resemble the learned noise distribution but misses other patterns, e.g., textural anomalies (see Figure 3). SI-VAE (Daniel and Tamar, 2021) learns the training distribution and provides superior pseudo-healthy reconstructions to baselines. However, it performs poorly in segmenting anomalies due to substantial false positive residual errors. $RA$ further improves the learned representations and improves the segmentation accuracy by 137% and 59% compared to SI-VAE and the best baseline, respectively.

### 4.2. Anomaly Detection

In this section, we assess the ability to detect global anomalies and artefacts as well as localize various pathologies. For the sake of comprehensiveness, we also compare our approach to the work of Salehi et al. (2021), which utilizes teacher-student methods and multi-scale knowledge distillation (MKD) for industrial and medical anomaly segmentation. Teacher-student networks face unique difficulties in medical anomaly segmentation, particularly in brain MRI, due to small datasets, subtle anomalies, and domain shifts.

**Global Detection.** The standard method for evaluating unsupervised anomaly detection is to compare the anomaly scores for unseen normal and pathological images. Table 2(a) shows the results for detecting global pathologies like small vessel chronic white matter ischemic changes and global artefacts like motion. With an AUROC score of 89.4, $RA$ outperforms the SOTA (DAE) by 54.4% and the best baseline (SI-VAE) by 22.3% on average.

**Pathology Localization.** In contrast to global detection, there is no standard for quantifying the performance of unsupervised pathology localization using ground truth bounding-boxes. From the clinical standpoint, we are interested in counting the amount of detected lesions while also limiting the number of false positives to reduce the burden on radiologists. As a result, we set the anomaly scores to produce no more than 5% false positives on normal samples and consider a detection to be true positive (TP) if it tags at least 10% of the pixels in the ground truth bounding box as anomalous. To assess the misclassifications, we calculate the false positives (FP) as the ratio of false pixel detection on healthy tissue

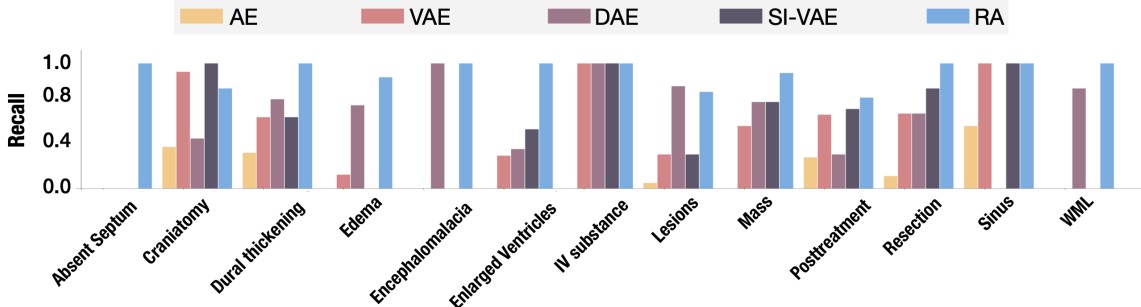

(*a*) Disease-specific detection. The majority of baselines fall short in identifying various diseases, including absent septum pellucidum, edema, encephalomalacia, or white matter lesions (WML). Moreover, even SOTA AEs struggle with structural anomalies like atrophy, where $RA$ improves the detection results to 100% compared to just 31% for DAE. $RA$ achieves nearly perfect recall, with the most errors occurring when missing small post-treatment changes (12/44) and small lesions (5/22).

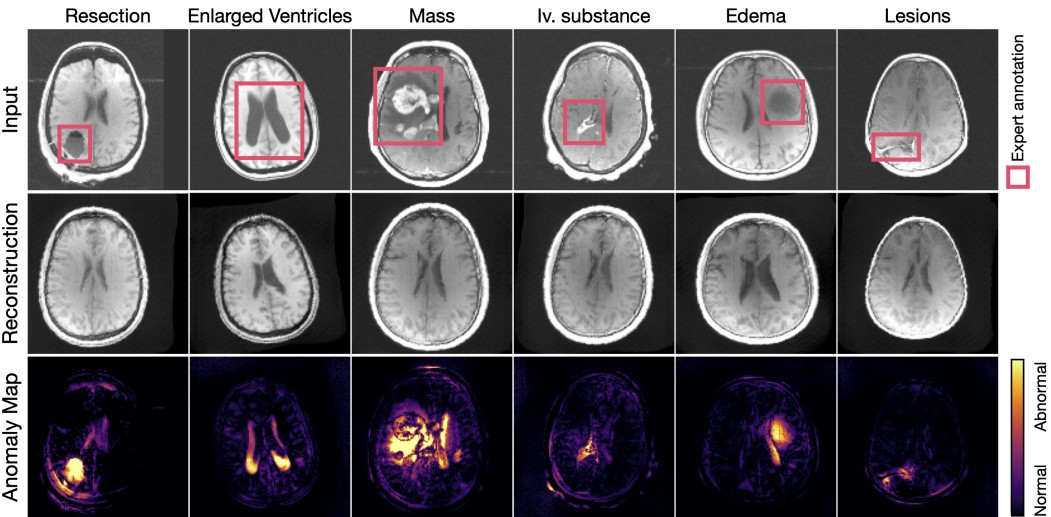

(*b*) $RA$ produces pseudo-healthy reconstructions allowing for the unbiased detection and precise localization of various intensity-based and structural pathologies.

Figure 4: Pathology Localization. Figure 4(*a*) shows quantitative results for different disease classes and Figure 4(*b*) shows qualitative examples of our method.

to true detection on anomalous tissue, which is given by: $FP = \sum_{h=0}^{H} s_h / \sum_{a=0}^{A} s_a$, with $H, A$ being the healthy and anomalous pixels, respectively. The overall performance is given by the average F1 score: $\overline{F1} = 1/N \sum_{i=0}^{N} (2 \times P \times TP)/(P + TP)$, with the precision $P = TP/(TP + FP)$. Table 2(*b*) shows numerical results. $RA$ detects 86.0% of diseases, whereas the SOTA DAE detects just over half of the pathologies (52.6%). The results for each disease class are shown in Figure 4(*a*). It should be noted that most methods fail to detect various types of anomalies, such as absent septum pellucidum, edema, encephalomalacia, or white matter lesions (WML). Moreover, even SOTA AEs struggle with

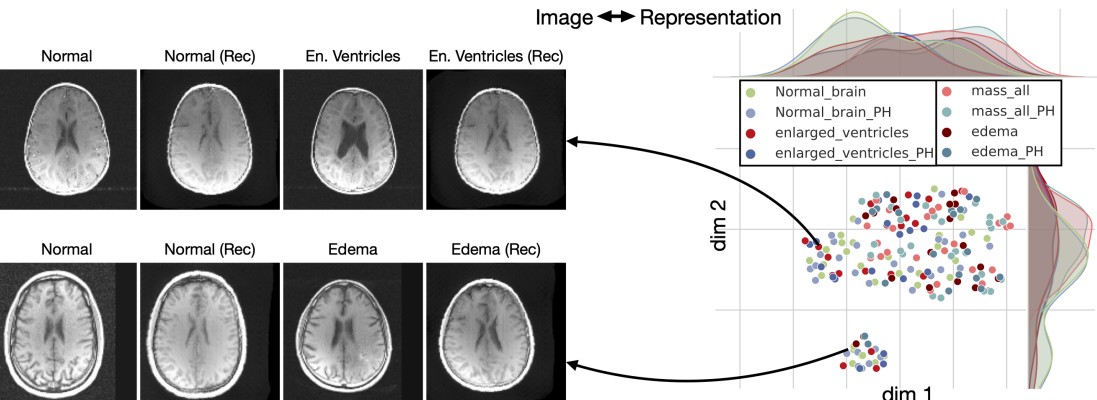

Figure 5: UMAP visualistion of latent representations extracted with $RA$ for normal, pathological, and pseudo-healthy inputs. On the right, two distinct clusters representing different scanners can be seen, with normal and abnormal samples in close proximity. On the left, sample inputs are displayed to highlight the differences.

structural pathologies, such as brain atrophy. For example, with a F1 score of 22.9, DAE detects only 31% of cases of enlarged ventricles. $RA$, on the other hand, detects all cases of enlarged ventricles with a F1 score of 79.7. $RA$ achieves nearly perfect recall on all diseases, with the most errors occurring when small post-treatment changes (12 out of 44) or small lesions (5 out of 22) are missed. Figure 4(b) shows qualitative examples of $RA$. Our method generates pseudo-healthy reconstructions of abnormal scans, allowing for the unbiased detection and localization of various diseases and paving the way for unsupervised pathology screening.

**Ablation: Effect of individual components.** The reverse loss improved the detection results, increasing the number of detected pathologies from 128/171 (F1:31.8) to 147/171 (F1:37.9). It especially improved the detection of subtle anomalies such as edema and small tumors to 16/18 and 24/26 from 9/18, 20/26, respectively. Histogram equalization mitigates the bias of not detecting subtle anomalies in terms of intensity differences, such as edema. Without it, edema detection decreases from 16/18 (F1:51.5) to 11/18 (F1: 26.4). Please refer to section D in Appendix for the full ablation results.

**Ablation: On anomaly maps.** It is challenging to compute useful anomaly scores. While pixel-wise residual errors are straightforward and provide precise anomaly maps, they strongly depend on the underlying intensities and are sensitive to small shifts (Meissen et al., 2022b). Perceptual anomaly maps (Zhang et al., 2018) have been investigated for related computer vision applications, e.g., industrial defect detection (Roth et al., 2022). They provide anomaly maps that correspond more closely to human perception. However, they have poor spatial resolution and use embeddings of networks pre-trained on natural images like ImageNet, which may not translate well to the medical imaging domain. Other approaches have tried to compute the likelihood of anomalies in the latent representation space (Pinaya et al., 2021; Pidhorskyi et al., 2018; Zimmerer et al., 2019a). For example, Zimmerer et al. (2019a) uses the KL divergence to distinguish between normal and abnormal samples. Figure 5 shows the umap (McInnes et al., 2018) visualization of latent embeddings of $RA$. It shows representations of normal, pathological and pseudo-healthy inputs. The normal and abnormal samples are close together within two distinct clusters

representing different scanners, as shown on the left. Thus, AEs to extract meaningful representations that only describe healthy populations, the latent embeddings will not provide important clues for detecting anomalies. Nonetheless, improving and generalizing anomaly scores is an important and exciting topic for future research.

## 5. Discussion

In this work, we have investigated whether AEs can learn meaningful representations and detect various pathologies on brain T1w MRIs. We have shown that standard, variational, recent adversarial and SOTA denoising AEs fail to detect multiple classes of non-hyperintense pathologies. We proposed novel *reversed AEs (RA)* to learn compact and detailed representations of the healthy distribution and improved the global and local pathology detection by 54.4% and 63.5% over SOTA DAE. In particular, *RA* considerably improved the detection of enlarged ventricles, achieving total recall and a F1 score of 79.7 (248% improvement). These findings pave the way for generalizing beyond hyperintensity thresholding to the unbiased detection of various diseases.

## 6. Acknowledgements

C.I.B. is in part supported by the Helmholtz Association under the joint research school "Munich School for Data Science - MUDS".

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

## Appendix A. Implementation details

We developed our deep learning framework using Python 3.8. All the methods were implemented in PyTorch. We ran the experiments on NVIDIA RTX A6000 and NVIDIA Tesla V100 GPUs. In the following we describe the implementation details of the baselines.

**Spatial AE (Baur et al., 2021).** We followed the official public implementation[1]. We trained the networks with a batch size of 8 using an ADAM optimizer with a learning rate of $5e-04$ and $L1$ reconstruction loss for up to 1500 epochs (early stopping after 500).

**VAE (Zimmerer et al., 2019a).** We followed the official public implementation[2]. We adapt the networks to the higher resolution ($128 \times 128$) of the brain MRI anomaly experiments by duplicating the last layer with 256 filters. We trained the networks with a batch size of 8, a latent dimensionality of 512 using an ADAM optimizer with a learning rate of $5e-04$ with a plateau learning rate scheduler (multiplies with 0.1 on plateaus), early stopping after 500 epochs and $L2$ reconstruction loss for up to 1500 epochs.

**DAE (Kascenas et al., 2022).** We followed the official public implementation[3]. We trained the network with a batch size of 8 using an ADAM optimizer with a learning rate of $1e-04$ and $L1$ reconstruction loss. We set the noise parameters to the default resolution (16) and std (0.2). We trained for up to 1500 epochs and early stopping after 500 consecutive epochs with no improvement greater than $1e-08$.

---

1. https://github.com/StefanDenn3r/Unsupervised_Anomaly_Detection_Brain_MRI
2. https://github.com/MIC-DKFZ/vae-anomaly-experiments/
3. https://github.com/AntanasKascenas/DenoisingAE/

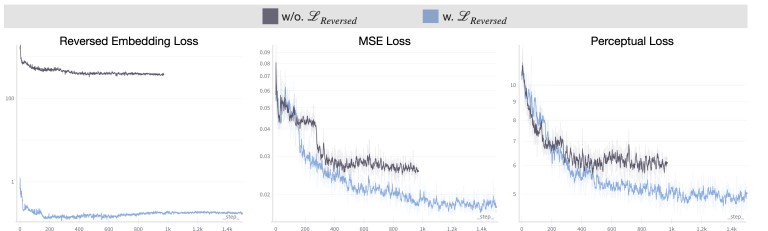
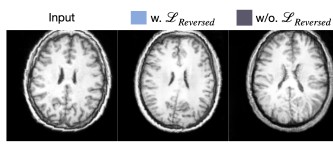

Figure 6: The reversed embedding loss improves the MSE and perceptual similarity on the healthy validation set and produces more accurate reconstructions.

**Soft-Intro VAE (SI-VAE) (Daniel and Tamar, 2021).** We followed the official public implementation[4]. We used 5 layers (64,128,256,512,512) with a bottleneck of 256 filters and a final sigmoid activation for the brain pathology experiments. We set $\beta_{KL} = 1, \beta_{Rec} = 1, \beta_{Neg} = 256$ and used ADAM optimizer with learning rates of $5e{-}05$ and batch size of 8. We trained for up to 1500 epochs (early stopping after 500 epochs).

## Appendix B. Training details

We trained the networks to reconstruct only healthy brains on 712 middle axial T1w brain MRI slices (131 FastMRI+, 581 IXI) and used 15 middle axial slices from the FASTMRI+ dataset for validation. We normalized the brain scans with the 98th percentile of the tissue intensities, padded them to be square and resized all scans to a resolution of $128 \times 128$. We used affine augmentations with a random rotation up to 10 degrees, up to 0.1 translation, scaling from 0.9 to 1.1, and horizontal flips with the probability of 0.5. Figure 6 shows the effect of the reversed embedding loss.

## Appendix C. Evaluation details

We performed all evaluation protocols on the FASTMRI+ dataset.

### C.1. Synthetic experiments

For the synthetic experiments in subsection 4.1 we altered 30 unseen, normal T1w MRI middle axial slices with synthetic anomalies. We used the ground truth normal images and generated anomaly masks to measure the pseudo-healthy reconstruction performance within the anomalous regions (MSE and LPIPS) and segmentation performance (AUPRC). **Intensity-based anomalies**. We augmented the normal images with three different ellipses. We uniformly sampled the three intensities to be: hypointense (1st-5h percentile), mean-ranged (30th-70th percentile), and hyperintense (95th-99th percentile). We chose the lengths of the ellipse axes to be normally distributed between 8 and 12, and the center coordinates in the range 32 to 96, with a random angle between 0 and 360 degrees. **Copy-Paste textural anomalies**. We first draw random rectangles( width and height within the range $25 - 50$). Then, we randomly select source and destination locations in the range of 25 to 128 - weight and 25 to 128 - height for x and y coordinates, respectively.

---

4. https://taldatech.github.io/soft-intro-vae-web/

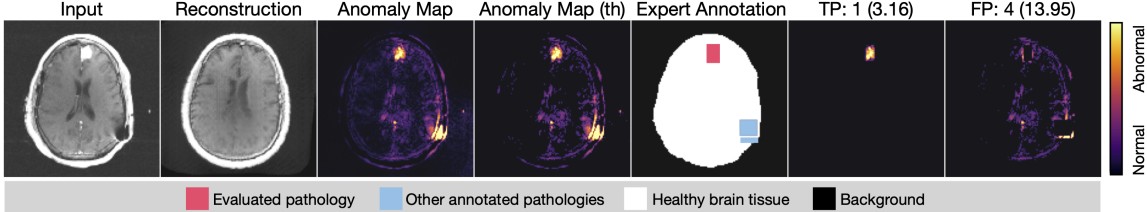

Figure 7: Pathology localization evaluation. From left to right: Input image, pseudo-healthy reconstruction, anomaly map, thresholded anomaly map, the expert annotation with the currently evaluated pathology, other annotated pathology, healthy tissue and background, the true positives map (sum: 3.16) and the amount of false positives (4 = 13.95 / 3.16).

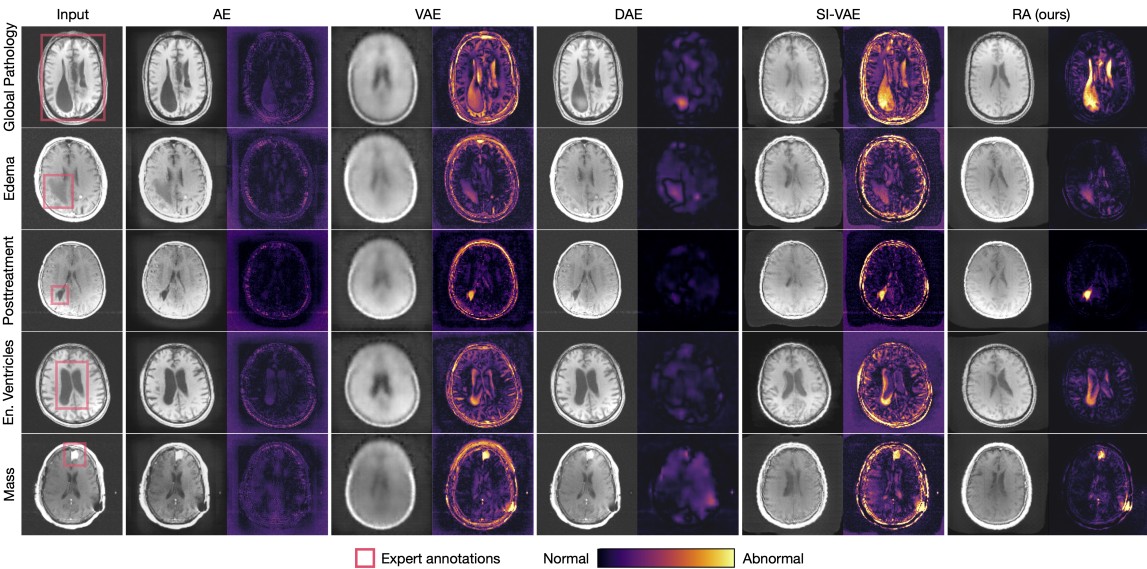

Figure 8: Global and Local Pathology Detection.

## C.2. Global Anomaly Detection

For the global anomaly detection evaluation, we used 30 normal, unseen middle axial slices, 27 global pathology scans (25 small vessel chronic white matter ischemic change, 1 global colpocephaly, 1 global extra-axial collection) and 23 global artefacts (9 motion artefacts, 14 tagged as 'possible artefact').

## C.3. Pathology Localization

We evaluate the localization of following pathologies: absent septum pellucidum (n=1), craniotomy (n=15), dural thickening (n=7), edema (n=19), encephalomalacia (n=1), enlarged ventricles (n=19), intraventricular substance (n=1), lesions (n=22), mass (n=26), posttreatment changes (n=44), resection (n=10), paranasal sinus opacification (n=2), and white matter lesions (n=5). Figure 7 shows how the localization of pathologies is quanti-

Table 3: Effect on individual components on the performance of $RA$.

(a) **Anomaly Detection.**

| Method | Average | Global Path. AUROC ↑ | Global Art. |
|---|---|---|---|
| RA (ours) | **89.4** | **89.0** | **89.8** |
| w/o Reverse Loss | 87.4 | 88.5 | 86.3 |
| w/o Hist. Eq. | 85.2 | 85.9 | 84.6 |
| w/o PL | 75.5 | 71.4 | 79.6 |
| w/o Hist. Eq. & w/o PL | 74.4 | 77.4 | 71.5 |

(b) **Pathology localization.**

| Method | Local Pathology #Detected ↑ | $\overline{F1}$(%) ↑ |
|---|---|---|
| RA (ours) | **147/171** | **37.9** |
| w/o Reverse Loss | 128/171 | 31.8 |
| w/o Hist. Eq. | 137/171 | 24.8 |
| w/o PL | 120/171 | 14.4 |
| w/o Hist. Eq & w/o PL | 105/171 | 14.4 |

fied. The anomaly map is first thresholded. The threshold is computed to produce at most 5% false positives on the 30 normal scans. We use the ground truth bounding box of the currently evaluated pathology (red) and calculate the amount of detected pixels inside the box ($> 10\% \rightarrow$ TP) and their sum (3.16). The amount of false positives is given by the ratio of false detections on healthy tissue (white) to the amount of correct detections inside the bounding box (red): $13.95/3.16 = 4$. Note that, most of the false positives in this case are around the annotated bounding boxes and for unlabeled artefacts (white dot below the ventricles). Figure 8 shows the qualitative comparison of all methods on detecting global and local pathologies.

## Appendix D. Effect of individual components

Table 3 shows the performance of each individual component for the experiments in subsection 4.2. The proposed **perceptual scores** counteract false positive detection and have the highest impact on the performance. The **histogram equalization** is especially beneficial in detecting subtle anomalies, e.g., it improved the edema detection from 4/18; F1:8.9 to 9/18; F1:16.2 without using a perceptual loss (PL); and from 11/18; F1: 26.4 to 16/18; F1:51.4 when combined with PL. The **reverse loss** improved the detection results, increasing the number of detected pathologies from 128/171; F1:31.8 to 147/171; F1:37.9 with our proposed anomaly map computation and from 82/171; F1:9.9 to 105/171; F1:14.4 when using simple residual differences.

