# OpenReview forum: "Generalizing Unsupervised Anomaly Detection: Towards Unbiased Pathology Screening"
_MIDL.io/2023/Conference — MIDL 2023 Oral_

### Official Review · Reviewer_Y42Z · 2023-01-26

**Confidence:** 4
**Preliminary Rating:** 4
**Recommendation:** Poster

**Summary:**

The authors proposed a new method in anomaly detection for MRI brain images based on auto-encoders, and explained their methodology with their revesred auto-encoder framework.
The authors compared rsults from their new reversed auto-encoder to other methodologies in auto-encoders in detection anomalies in MRI brain.
The authors showed superior results from the reversed auto-encoder especially in mass, non-hyperintense lesions.

**Strengths:**

The strength of the paper is taht the method was an improvement on typical auto-encoder framework and generated interesting results in MRI brain demonstrating its improvments in experimental data, and also this is an area of clinical unmet need as well.

**Weaknesses:**

A more detailed explanation of the methodology could be helpful especially in anomaly detection between pathological and pseudo healthy reconstruction and a discussion on how different ways of calculation anomaly compared to ground truth could be helpful.

Multiple coexisting anomalies might be considered as well.

**Deanonymize Review:**

no

**Paper Type:**

both

**Questions To Address In The Rebuttal:**

Refined discussion around weakness points mentioned above
Also might add explanations on why choose only one view of each MRI and potential benefits and downsides of including more than one view per MRI especially given most lesions show in more than one view and image

---

### Official Review · Reviewer_nWFd · 2023-02-02

**Confidence:** 4
**Preliminary Rating:** 4
**Recommendation:** Poster

**Summary:**

In this paper, the authors proposed a novel method for anomaly detection on brain MRI scans. Existing methods, such as VAE and DAE, either suffer from blurry reconstruction or reconstruct the anomalies as well resulting in poor detection performance, especially for non-hyperintense anomalies. While soft-introVAE soft-introVAE could produce many false positives in the detection, the authors modified the original soft-introVAE and proposed reversed autoencoder (RA), by adding a loss that enforces similarity at each encoding layer. To localize the anomalies, existing methods mostly make use of the reconstruction loss, which can be too dependent on the intensity values. The authors proposed a modified calculation using perceptual loss to detect the anomalies.

**Strengths:**

The method is overall clear and easy to follow, the conclusions are supported by sufficient evaluation and compared to SOTA methods.

Specifically, by adding the similarity loss at each encoding layer, RA is able to reconstruct the pseudo-healthy image more faithfully as shown in Fig.3 and Fig.6. Together with the proposed anomaly score using perceptual loss, compared to existing methods, the proposed method localizes the anomalies more accurately with fewer false positives.

**Weaknesses:**

To use auto-encoding models for anomaly detection, one would hope that, during detection, the input with anomalies can be reconstructed as a normal image that bears most similarities to the original one except for the anomalous regions. However, most auto-encoding models are enforced explicitly to have such a behavior. It could make unclear the relation between improving reconstruction quality and improving anomaly detection. Could authors provide further insights on how reliable the auto-encoding model is to reconstruct an anomaly-free image while not changing healthy structures?

The proposed anomaly scores can take structural alteration into account. The score should be general and can be applied in the detection using other methods. Are the detection results from the other methods also calculated with this score or reconstruction? If it’s the latter, how much will the new score improve their detection accuracy? On the other hand, for the perceptual loss, how is it computed exactly? The notation for x_{ph} seems to be unexplained in the paper.

To compute anomaly scores, the authors also mentioned the use of adaptive histogram equalisation, how much will the scores differ with and without this operation?

As the authors mentioned the distillation method used in the work of Salehi et al, CVPR2021. Their method is also used for anomaly detection, is it possible to include it as well in the comparison for this work?

For Fig.4 and Fig.8, is it possible to show a pixel-wise annotation to better compare the detected anomalies and the ground truth?


**Deanonymize Review:**

no

**Paper Type:**

methodological development

**Questions To Address In The Rebuttal:**

- Please comment in the reliability of using auto-encoding methods for anomaly detection, specifically, is it guaranteed theoretically that such models can reconstruction an anomy-free image without altering non-anomalous regions in the original input?

- Is it possible to use the proposed anomaly scores for other methods, such as SI-VAE, will it improve their performance too?

- Please provide more details the calculation of the perceptual loss.

- Compare the scores with and without adaptive histogram equalisation.

- If possible, could the authors include the work by Salehi et al (CVPR2021) in the comparison?

- If possible, could the authors provide pixel-wise annotations for some figures for easier comparison? For example, in the first row of Fig.8, the annotation almost covers the whole image. As a non-expert, it's difficult to determine what pixels belong the anomalies by simple visual inspection.

---

### Official Review · Reviewer_9yCN · 2023-02-02

**Confidence:** 4
**Preliminary Rating:** 4
**Recommendation:** Poster

**Summary:**

This paper studies Autoencoders (AE) in the context of medical anomaly detection. While different improvements on the standard AE have been achieved, their limitations on anomaly detection have been exposed in this paper, particularly: problem in reconstruction due to inconsistent representations, and unsensitiveness to not intense anomalous regions. The authors then proposed the so-called Reversed Autoencoder (RA), which consists in a regularization approach on the encoder of a variational AE based on the similarity of the features, using an additional loss term. Experiments on two medical datasets were conducted, demonstrating the advantages of this method against other AE models.

**Strengths:**

This paper is well written, and clearly describes the limitations of AE models in anomaly detection, particularly in the context of medical imaging. The authors propose a novel approach to reverse the output of an AE by forcing the representations of a reconstructed image to be similar to the ones of the original image. By this, they achieve important improvements on Anomaly Detection on two different datasets and synthetic problems. The study is well structured and detailed, and the discussions are driven by the obtained data.

**Weaknesses:**

Although the reversed similarity measurement of representations is novel in the context of AEs, similar losses have been used in different image analysis applications (see reference below). This somewhat reduces the novelty of the method.

The method demonstrates good performance, specially compared to other AE models, but there is a lack of comparison with state-of-art
anomaly detection methods, for instance based on normalizing flows or student-teacher distillation.

Experiments also lack a proper ablation study evaluating the direct contribution of the reverse loss and histogram equalization strategy while keep all other components the same (e.g., same architecture and data augmentation).

Oktay, Ozan, Enzo Ferrante, Konstantinos Kamnitsas, Mattias Heinrich, Wenjia Bai, Jose Caballero, Stuart A. Cook et al. "Anatomically constrained neural networks (ACNNs): application to cardiac image enhancement and segmentation." IEEE transactions on medical imaging 37, no. 2 (2017): 384-395.

**Deanonymize Review:**

no

**Detailed Comments:**

* The proposed method uses a similarity metric as a loss function. This loss component measures how similar the representation for the original image and the reconstructed image are. Does this extra computation cause an important computational overhead during training?

* The regularization method based on the similarity of the representations is based on the work by Salchi et al. (2021), and is composed by a cosine similarity term and a MSE score. What is the effect of each of these related components ? Furthermore, are both needed to measure similarity? An ablation study on the importance of these terms could be useful.

* These new RA models outperforms other standard AE techniques. How does this approach compare with other Anomaly Detection methods? Some more experimental comparison can help visualizing where this method stands in terms of other AD strategies.

* While the method does a good job at spotting anomalies, it also reconstructs brains which are very different from the original ones (e.g., different cortical folds or different shape of ventricles). For example, in the first column of 4(b), the frontal region of brain is completely changed by the reconstruction. This could lead to many false positives or impede the detection of more subtle anomalies (small, low contrast legions). Please comment.

* Eq (2): I believe this objective should be maximized (it is not a loss)

* In Section 4.2, why use such a low percentage of pixels (10%) to define the true positives?




**Paper Type:**

methodological development

**Questions To Address In The Rebuttal:**

The main issues to address in the rebuttal are to 1) better emphasize the novelty and value of the contributions; 2) measure (or at least explain) the impact of the reverse loss and histogram equalization on the method using the same setting; 3) if possible, compare against state-of-art methods for anomaly detection. See detailed comments for less critical points.

---

### Meta-Review · Area_Chair_YW4g · 2023-02-20

**Recommendation:** Accept (Poster)
**Confidence:** 5

**Metareview:**

Despite the initial borderline scores, the authors addressed satisfactorily most concerns raised by the reviewers, providing detailed and constructive responses, and updating the manuscript accordingly. I think this work could lead to interesting discussions in the conference and thus recommend its acceptance.